

# Anomalous dimensions of potential top-partners

**Diogo Buarque Franzosi⋆ and Gabriele Ferretti**

Department of Physics, Chalmers University of Technology,
Fysikgården 1, 41296 Göteborg, Sweden

⋆ buarque@chalmers.se

## Abstract

We discuss anomalous dimensions of top-partner candidates in theories of Partial Compositeness. First, we revisit, confirm and extend the computation by DeGrand and Shamir of anomalous dimensions of fermionic trilinears. We present general results applicable to all matter representations and to composite operators of any allowed spin. We then ask the question of whether it is reasonable to expect some models to have composite operators of sufficiently large anomalous dimension to serve as top-partners. While this question can be answered conclusively only by lattice gauge theory, within perturbation theory we find that such values could well occur for some specific models. In the Appendix we collect a number of practical group theory results for fourth-order invariants of general interest in gauge theories with many irreducible representations of fermions.

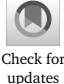

## 1 Introduction

All models of physics beyond the standard model attempting to explain the origin of the electro-weak (EW) scale face a fundamental tension. On the one hand, they need to have

additional particles or phenomena near that scale, while, on the other, they must preserve the stringent constraints from flavor-changing, CP-violating processes etc. In the context of strongly-coupled solutions, this generically requires a decoupling of the EW scale from the "flavor" scale $\Lambda_F$ where these new effects come into play. In order not to throw the baby out with the bathwater, the baby being the top quark mass, some operators must acquire a large anomalous dimension to survive the long journey from the flavor scale to the EW scale. This fact is common in all attempts, such as walking technicolor, conformal technicolor, holography and partial compositeness.

Here we consider the particular case of partial compositeness [1] (see [2, 3] for reviews) realized via a four-dimensional gauge theory with fermionic matter in the spirit of [4, 5]. In [6, 7] the set of potential models was narrowed down from the full list in [5] to a list of twelve most promising one (containing the original [4]). There is also the attempt to use a QCD-like theory for these purposes [8].

Although the model must obviously be confining in the IR, it may start *inside* the conformal window and rely on some relevant deformation (like a fermion mass) to leave the fixed point at parametrically low scales, triggering confinement [9], (see also [8] and [6]). What is important is that, after this happens, there are enough light fermions left to guarantee a sensible phenomenology. The interest is thus to look at confining models adjacent to conformal models with large anomalous dimensions. The actual number of dynamical fermions, and thus whether the model is in the conformal window or not, depends on the relation between the masses and the energy scale.

Without reviewing the construction, which is discussed in detail in the above papers, suffices to say that each of these models consist of a unitary or symplectic gauge group (hypercolor) with fermions (hyper-quarks) in the fundamental **F** and antisymmetric $\mathbf{A}_2$ irreducible representation (irrep) or of an orthogonal hyper-color group with hyper-quarks in the fundamental **F** and spinorial **Spin** irrep. These models have the advantage of being amenable to lattice studies and, indeed, work has been done in the unitary and symplectic case by the groups [10–14] and [15, 16] respectively.

In particular, one of the models in the list [6, 7], based on the gauge group $SU(4)$ and spelled out in more details in [17], has been put under intense scrutiny, albeit with a smaller number of hyper-quarks than those required for applications to EW breaking (4 v.s. 5 Majorana hyper-quarks in the $\mathbf{A}_2$ and 2 v.s. 3 Dirac hyper-quarks in the $(\mathbf{F}, \overline{\mathbf{F}})$).

The first important lattice result [11] concerning the $SU(4)$ model, was to show that in the chiral limit (massless hyper-quarks) the mass of the potential top-partners ("chimera baryons" in their language) is not the smallest among the non-pseudo-Nambu–Goldstone states, but is in fact slightly higher than that of the vector resonances, with a mass of roughly $M_T \approx 8.5f$, where, in the notation of [17], $v = f \sin(\langle h \rangle / f) = 246$ GeV [1]. EW precision tests require the fine-tuning parameter $v^2/f^2 < 0.1$ and this puts the top-partners in this model out of reach of the LHC, ($M_T > 6.8$ TeV) if one assumes that the result can be extrapolated to a more realistic number of fermions.

The second, more recent, result [12] concerns the mass of the top quark or, equivalently, its Yukawa coupling $y_t$. Assume that the theory enters a conformal regime between the "flavor" scale $\Lambda_F \gtrsim 10^4$ TeV and the hyper-color confinement scale $\Lambda_{\text{HC}} \lesssim 10$ TeV where the fermionic trilinear composite operator $\mathcal{O}$, representing the top-partner, has scaling dimension $\Delta = 9/2 + \gamma^*$. Under some specific assumptions, [12] shows that at the scale $\Lambda_{\text{HC}}$ the Yukawa

---

[1] [11] finds $M_T \approx 6.0F_6$, having defined $F_6 = \sqrt{2}f$.

coupling turns out to be [2]

$$y_t \approx 0.06 \left( \frac{g_F^2}{\Lambda_F^2} \left( \frac{\Lambda_{\mathrm{HC}}}{\Lambda_F} \right)^{\gamma^*} \right)^2 f^4. \qquad (1)$$

The small coefficient in front of (1) is problematic, since we need $y_t \approx 1$. To overcome this problem requires $-3 < \gamma^* < -2$, the lower bound being required by unitarity. Notice however that even a larger value for the coefficient would require similarly strong renormalization effects, $\gamma^* \approx -2$.

To assess the viability of models of this type it is thus necessary to understand where the edge of the conformal window for such theories lie and what the anomalous dimensions of the fermionic operators at the edge might be. Both of these issues can only be truly answered by strong coupling techniques, such as lattice gauge theory. In this paper we content ourselves with performing various perturbative computations.

We start by revisiting the results of [18] and extending them to all other relevant cases by using the convenient Weyl formalism, used in [19] for baryons in QCD. As for the search of a fixed point, we are forced to be more qualitative, but we use the state of the art four-loop $\beta$-function for generic gauge theories with multiple fermionic irreps of Zoller [20].

Having stated up-front that a perturbative analysis will never be able to quantitatively answer the question of phenomenological interest, what is the use of doing it? In our opinion, the main reason is to guide us towards the most promising models, and to qualitatively assess the likelihood that such large anomalous dimensions might be realized. As an extreme case, imagine comparing two theories, one that has a positive one-loop $\gamma$-function and one that has a negative one. Clearly, given the need to have $\gamma^* < -2$ at the fixed point, the second one will make a more promising candidate for a non perturbative analysis. Similar heuristic considerations can be made about the existence of fixed points and their relative strength. Given the amount of effort required to perform a lattice calculation, such small hints can be valuable.

The paper is organized as follows: In Section 2 we present the computation of the one-loop $\gamma$-function in full generality using the Weyl spinor formalism. This generalizes the results of [18] to all possible models.

In Section 3 we try to estimate the edge of the conformal window. We use various heuristic arguments such as stability considerations and the proposed criteria of [21–23]. We apply the results to the models of phenomenological interest denoted M1...M12 in [7]. We compare the $\gamma$-functions of the various operators in the models and estimate the numerical values of the anomalous dimensions of those corresponding to potential top-partners.

Section 4 contains our conclusions where we try to present a balanced view of the situation regarding these issues.

The Appendix contains the group theory results that are needed for the numerical evaluation of the four-loop $\beta$-function [20]. At fourth order one needs to consider the fourth order Casimir operators and also the mixed product of the fourth-order invariant tensors between different irreps. We tabulate these values for the smallest irreps of each Lie algebra. These results can be useful for other applications as well and the Appendix can be read quite separately from the rest of the paper.

---

[2] From [12], this number comes about as $((0.3)^2/6) \times 4$, where 0.3 and 6 are the overlap functions $Z$ and the top-partner's mass in units of $F_6$ and the factor 4 is the rescaling $F_6^4 = 4f^4$. We also point out that formulas of this type are sensitive to the details of the UV mechanisms generating the couplings.

## 2  The one-loop $\gamma$-function

Our first goal is to compute the one-loop $\gamma(g)$ function for the trilinear operators of interest. We use dimensional regularization and work in the Feynman gauge. In asymptotically free theories, this function is sometimes referred to as the "anomalous dimension" of the operator, although in a CFT the true anomalous dimension is the value $\gamma^*$ that the function assumes at the fixed point $g^*$.

The operators of interest are objects of the type $\langle X_\alpha Y_\beta Z_\gamma \rangle$ and $\langle X_\alpha Y^\dagger_{\dot\beta} Z^\dagger_{\dot\gamma} \rangle$, where $X, Y, Z$ are three generic Weyl fermions of the hyper-color gauge group $G_{\mathrm{HC}}$ and $\langle \ldots \rangle$ denotes a $G_{\mathrm{HC}}$ invariant combination. Dotted and undotted indices denote right- and left-handed spinors respectively. Operators with an odd number of dotted indices can be obtained by parity conjugation and have the same anomalous dimension since they can be combined into a composite Dirac spinor.

The operator $\langle X_\alpha Y_\beta Z_\gamma \rangle$ can be further decomposed into a $(s_L, s_R) = (3/2, 0)$ Lorentz irrep, by fully symmetrizing the Weyl indices, and two irreps $(1/2, 0)$ and $(1/2, 0)'$, possibly mixing with each-other, while the operator $\langle X_\alpha Y^\dagger_{\dot\beta} Z^\dagger_{\dot\gamma} \rangle$ decomposes into a $(1/2, 1)$ and a $(1/2, 0)''$ Lorentz irrep. Operators carrying different spin or different unbroken flavor symmetries do not mix with each-other. For this last reason, the $(1/2, 0)''$ irrep does not mix with the two previous ones.

We need to address a couple of issues about operator mixing that are not relevant for applications to partial compositeness. The first issue arises whenever there is more than one $G_{\mathrm{HC}}$ singlet in the decomposition of $R_X \otimes R_Y \otimes R_Z$, where $R_X$ denotes the $G_{\mathrm{HC}}$ irrep under which $X_\alpha$ transforms and so on. Operators of such kind would mix, but luckily they never occur in models of partial compositeness, as can be easily checked.

Another issue arises when one of the three fermions transforms in the adjoint of $G_{\mathrm{HC}}$, say $R_X = \mathbf{Ad}$, and the remaining two combine into a singlet of the *flavor* symmetry. This kind of operator can mix with a different one, schematically denoted by $\langle DFX \rangle$, where $F$ is the $G_{\mathrm{HC}}$ field strength and $D$ is the covariant derivative needed to have classical dimension 9/2. However, after decomposing this operator into irreps of the Lorentz group, one can show that the part of the operator that mixes can be removed by field re-definition using the equations of motion (as explained for QCD in e.g. [24]). Recall that $F$ can be split into self-dual and anti-self-dual components $f_{\alpha\beta}$ and $\bar{f}_{\dot\alpha\dot\beta}$ in Weyl notation. When acting on e.g. $f_{\alpha\beta}$ by a covariant derivative $D_{\gamma\dot\alpha} = \sigma^\mu_{\gamma\dot\alpha} D_\mu$ we obtain a tensor with three undotted and one dotted index. The irreducible component (the one that cannot be re-written by using the equation of motion) is obtained by fully symmetrizing in the undotted indices and can be denoted by $Df_{\alpha\beta\gamma\dot\alpha} \in (3/2, 1/2)$ (equivalently $D\bar{f}_{\alpha\dot\alpha\dot\beta\dot\gamma} \in (1/2, 3/2)$). Combining with $X^\dagger_{\dot\delta}$ to have an even number of dotted indices yields $\langle Df_{\alpha\beta\gamma\dot\alpha} X^\dagger_{\dot\delta} \rangle \in (3/2, 0) \oplus (3/2, 1)$ and $\langle D\bar{f}_{\alpha\dot\alpha\dot\beta\dot\gamma} X^\dagger_{\dot\delta} \rangle \in (1/2, 1) \oplus (1/2, 2)$, none of which can interfere with the renormalization of the putative top-partner.

Thus, we bypass both these unnecessary complications by considering operators made out of three *distinct* fermions for which there is a *unique* $G_{\mathrm{HC}}$ invariant. Some of the fermions may well transform under the same irrep of $G_{\mathrm{HC}}$ but the uniqueness is guaranteed by picking a different flavor index.

We are now ready to perform the computation. First of all, the wave-function renormalization for each Fermi field reads $X_{\mathrm{bare}} = Z_X^{1/2} X$, with $Z_X = 1 + \frac{g^2}{16\pi^2} \frac{1}{\epsilon} a_X$, $\epsilon = 4 - d$ and $a_X \equiv -2C_X$, $C_X$ being the (eigenvalue of the) quadratic Casimir of $R_X$ and similarly for $Y$ and $Z$[3].

We further need the composite operator renormalization which we write as $\langle XYZ \rangle^I_{\mathrm{finite}} = Z^{IJ} \langle XYZ \rangle^J$ where $I$ and $J$ run over the Lorentz irreps discussed above, including

---

[3]Recall that we work in the Feynman gauge $\xi = 1$. In general $a_X \equiv -2C_X \xi$.

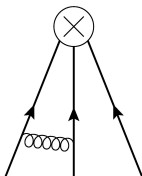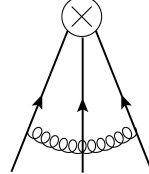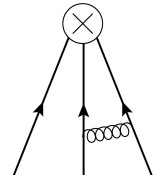

Figure 1: Diagrams giving the divergent part of the "$\mathcal{P} \times I$" vertex in the fully chiral (no dotted indices) case.

the ones with daggered fermions. In general $Z^{IJ}$ is a matrix, but, as we saw, at most a $2 \times 2$ block needs to be diagonalized. We write it as $Z^{IJ} = \delta^{IJ} + \frac{g^2}{16\pi^2} \frac{1}{\epsilon} a^{IJ}$. The one-loop $\gamma$-function is then given, in this notation, by

$$\gamma^{IJ}(g) = \frac{g^2}{16\pi^2}\left(a^{IJ} - \frac{1}{2}\delta^{IJ}(a_X + a_Y + a_Z)\right). \tag{2}$$

Thus all that remains is to compute the divergent part of $\langle XYZ \rangle^I$ in renormalized perturbation theory.

For this last step we need to be more specific and write the projection operators in spin space explicitly. We set

$$\langle X_{\alpha'}Y_{\beta'}Z_{\gamma'}\rangle_{(s_L,s_R)} = \mathcal{P}(s_L,s_R)^{\alpha\beta\gamma}_{\alpha'\beta'\gamma'}I_{xyz}X^x_\alpha Y^y_\beta Z^z_\gamma,$$

$$\langle X_{\alpha'}Y^\dagger_{\dot\beta'}Z^\dagger_{\dot\gamma'}\rangle_{(s_L,s_R)} = \mathcal{P}(s_L,s_R)^{\alpha\dot\beta\dot\gamma}_{\alpha'\dot\beta'\dot\gamma'}I_{x\bar y\bar z}X^x_\alpha Y^{\dagger\bar y}_{\dot\beta} Z^{\dagger\bar z}_{\dot\gamma}, \tag{3}$$

where $I_{xyz}$ and $I_{x\bar y\bar z}$ are the (unique) invariant tensors in the product of the respective irreps, $(x = 1\ldots\dim(R_X)$ and so on), and

$$\mathcal{P}(3/2,0)^{\alpha\beta\gamma}_{\alpha'\beta'\gamma'} = \frac{1}{6}\left(\delta^{\alpha\beta\gamma}_{\alpha'\beta'\gamma'} + \delta^{\gamma\alpha\beta}_{\alpha'\beta'\gamma'} + \delta^{\beta\gamma\alpha}_{\alpha'\beta'\gamma'} + \delta^{\beta\alpha\gamma}_{\alpha'\beta'\gamma'} + \delta^{\gamma\beta\alpha}_{\alpha'\beta'\gamma'} + \delta^{\alpha\gamma\beta}_{\alpha'\beta'\gamma'}\right),$$

$$\mathcal{P}(1/2,0)^{\alpha\beta\gamma}_{\alpha'\beta'\gamma'} = \frac{1}{4}\left(\delta^{\alpha\beta\gamma}_{\alpha'\beta'\gamma'} - \delta^{\gamma\beta\alpha}_{\alpha'\beta'\gamma'} + \delta^{\beta\alpha\gamma}_{\alpha'\beta'\gamma'} - \delta^{\beta\gamma\alpha}_{\alpha'\beta'\gamma'}\right),$$

$$\mathcal{P}'(1/2,0)^{\alpha\beta\gamma}_{\alpha'\beta'\gamma'} = \frac{1}{4}\left(\delta^{\alpha\beta\gamma}_{\alpha'\beta'\gamma'} - \delta^{\beta\alpha\gamma}_{\alpha'\beta'\gamma'} + \delta^{\gamma\beta\alpha}_{\alpha'\beta'\gamma'} - \delta^{\gamma\alpha\beta}_{\alpha'\beta'\gamma'}\right), \tag{4}$$

$$\mathcal{P}(1/2,1)^{\alpha\dot\beta\dot\gamma}_{\alpha'\dot\beta'\dot\gamma'} = \frac{1}{2}\left(\delta^{\alpha\dot\beta\dot\gamma}_{\alpha'\dot\beta'\dot\gamma'} + \delta^{\alpha\dot\gamma\dot\beta}_{\alpha'\dot\beta'\dot\gamma'}\right),$$

$$\mathcal{P}''(1/2,0)^{\alpha\dot\beta\dot\gamma}_{\alpha'\dot\beta'\dot\gamma'} = \frac{1}{2}\left(\delta^{\alpha\dot\beta\dot\gamma}_{\alpha'\dot\beta'\dot\gamma'} - \delta^{\alpha\dot\gamma\dot\beta}_{\alpha'\dot\beta'\dot\gamma'}\right).$$

The computation of the divergent part in Fig. 1 can thus be regarded as the renormalization of the vertex "$\mathcal{P} \times I$" and can be subdivided into a spin part, a gauge part and a simple loop-integral common to all diagrams, since the divergent part does not depend on the incoming momenta:

$$\int \frac{\mathrm{d}^d k}{(2\pi)^d} \frac{(p_i-k)^\mu(p_j+k)^\nu}{k^2(p_i-k)^2(p_j+k)^2} = -\frac{i}{32\pi^2}\frac{1}{\epsilon}\eta^{\mu\nu} + \text{ finite.} \tag{5}$$

For illustration purposes, we show the expression of the first diagram for the fully chiral (no daggered fermions) vertex displayed in Fig. 2 using the notation of [25]

$$\text{Diagram} = -ig^2(\sigma^\mu\bar\sigma_\nu)^\alpha_\delta\left(\sigma_\mu\bar\sigma_\rho\right)^\beta_\lambda \delta^\gamma_\eta \mathcal{P}(s_L,s_R)^{\alpha'\beta'\gamma'}_{\alpha\beta\gamma} \times$$

$$\left(I_{xyz}T^a(R_X)^x_{x'}T^a(R_Y)^y_{y'}\delta^z_{z'}\right) \times \int \frac{\mathrm{d}^d k}{(2\pi)^d}\frac{(p_1-k)^\nu(p_2+k)^\rho}{k^2(p_1-k)^2(p_2+k)^2}, \tag{6}$$

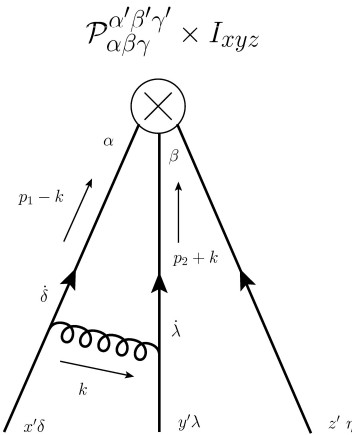

$$\mathcal{P}^{\alpha'\beta'\gamma'}_{\alpha\beta\gamma} \times I_{xyz}$$

Figure 2: Example of loop diagram giving rise to eq. (6).

with $T^a(R)$ denoting the generators of $G_{\mathrm{HC}}$ in the irrep $R$.

The spin algebra is a bit tedious but straightforward. Since we have gone to the trouble of performing the decomposition, it is now very convenient to simply pick one component in the spin multiplet by choosing some specific values for $\alpha'\beta'\gamma'$.

The gauge factor can be computed once and for all by the following observation. The invariant tensor $I_{xyz}$ can also be seen as the projector $R_X \otimes R_Y \to \bar{R}_Z$. Hence

$$(T^a(R_X) \otimes \mathbf{1} + \mathbf{1} \otimes T^a(R_Y))^2 \equiv C_X \otimes \mathbf{1} + 2T^a(R_X) \otimes T^a(R_Y) + \mathbf{1} \otimes C_Y \to C_Z, \qquad (7)$$

yielding, just like with the usual trick for adding angular momenta in quantum mechanics,

$$I_{xyz} T^a(R_X)^x_{x'} T^a(R_Y)^y_{y'} \delta^z_{z'} = \frac{1}{2}(C_Z - C_X - C_Y) I_{x'y'z'}. \qquad (8)$$

Putting it all together, after some algebra we obtain the following general expressions for the $a^{IJ}$ coefficients, valid under the two very mild restrictions mentioned at the beginning of this section.

$$
\begin{aligned}
a_{(1/2,0)} &= \begin{pmatrix} -6C_X - 2C_Y + 2C_Z & -4C_X + 4C_Y & 0 \\ 4C_Y - 4C_Z & 2C_X - 2C_Y - 6C_Z & 0 \\ 0 & 0 & 2C_X - 4C_Y - 4C_Z \end{pmatrix} \\
a_{(3/2,0)} &= 0 \\
a_{(1/2,1)} &= -2C_X.
\end{aligned}
\qquad (9)
$$

The block diagonal $3 \times 3$ matrix $a_{(1/2,0)}$, with components $a^{IJ}_{(1/2,0)}$ ($I, J = 1, 2, 3$), indicates the mixing between the operators with $\mathcal{P}(1/2,0)$, $\mathcal{P}'(1/2,0)$, with the last entry representing $\mathcal{P}''(1/2,0)$. The remaining coefficients are single numbers and we do not show any index.

Eq. (9) combined with (2) yields the final expression for the $\gamma$-functions, in the same

notation as (9):

$$\gamma_{(1/2,0)}(g) = \frac{g^2}{16\pi^2} \begin{pmatrix} -5C_X - C_Y + 3C_Z & -4C_X + 4C_Y & 0 \\ 4C_Y - 4C_Z & 3C_X - C_Y - 5C_Z & 0 \\ 0 & 0 & 3C_X - 3C_Y - 3C_Z \end{pmatrix}$$

$$\gamma_{(3/2,0)}(g) = \frac{g^2}{16\pi^2}(C_X + C_Y + C_Z) \tag{10}$$

$$\gamma_{(1/2,1)}(g) = \frac{g^2}{16\pi^2}(-C_X + C_Y + C_Z).$$

As expected, we see that the last $\mathcal{P}''(1/2, 0)$ component does not mix with the other two. In this general case, the diagonalization of $\gamma_{(1/2,0)}(g)$ yields a non-linear expression in the Casimirs due to the square-root of the discriminant of the characteristic polynomial $\Delta = 16(C_X^2 + C_Y^2 + C_Z^2 - C_X C_Y - C_X C_Z - C_Y C_Z)$. However, in all cases of interest, at least two of the three Casimirs are the same and this makes the discriminant into a perfect square, restoring linearity.

Before going forward, we better check that we reproduce the well known anomalous dimensions for baryon operators in QCD. Taking $G_{\text{HC}} = SU(3)$ and $C_X = C_Y = C_Z = C_{\mathbf{F}} = 4/3$ we obtain $\gamma_{(1/2,0)}(g) = \frac{4}{3}(-3)\frac{g^2}{16\pi^2}\mathbf{1}$, $\gamma_{(3/2,0)}(g) = \frac{4}{3}(+3)\frac{g^2}{16\pi^2}$ and $\gamma_{(1/2,1)}(g) = \frac{4}{3}(+1)\frac{g^2}{16\pi^2}$, in agreement with the computation of [19, 26]. (For these operators results are also available for two-loops [27, 28] and three-loops[4] [32].)

A more stringent check is to reproduce the results of [18], which are directly relevant for partial compositeness. They have computed the $\gamma$-functions for the $(1/2, 0)$ operators in a $SU(4)$ gauge theory with two fields in the fundamental and one in the anti-symmetric and for a $SO(2n)$ theory with one fundamental and two spinor irreps. These numbers also match, as it is shown in the next section, by comparing [18] with Table 2.

More specifically, the numbers for $SO(2n)$ match up to an overall factor of 4 but this is not an inconsistency and it is simply due to a different normalization of the generators. The same normalization affects the $\beta$-function and cancels out in the physical (scheme-independent) value of $\gamma^*$.

## 3  Applications to Partial Compositeness

We are now in the position of applying the results of the previous section to models that are of interest to partial compositeness. The candidate models of Partial Compositeness we are interested in are summarized in Table 1. They were selected [6, 7] from a much longer list [5] as the most promising ones after imposing a certain amount of criteria that we shall not review here.

By choosing $X, Y, Z$ to be either $\psi$ or $\chi$ or, for complex irreps, their charge conjugates, one can obtain the expressions for the respective $\gamma$-functions to one-loop. In Table 2 we present the full list of coefficients $A$ for the twelve models in Table 1, with the understanding that[5]

$$\gamma(g) = \frac{g^2}{16\pi^2}A. \tag{11}$$

---

[4] See [29] and [30] for a clarification about the sign convention and a factor of 2 discrepancy in the overall normalization, also relevant for [31].

[5] Although this is unlikely to have caused any trouble, we feel compelled to mention that the preliminary results presented by one of us (GF) at a few recent seminars used a different sign convention and incorrectly stated some of the results for the $(3/2, 0)$ case.

Table 1: The gauge and matter content of the models of interest for Partial Compositeness. The seemingly haphazard ordering is due to the fact that they were labeled following the cosets they give rise to (not shown here). **Spin** denotes the spinorial representation of $SO(N)$, $\mathbf{A}_2$ and $\mathbf{F}$ denote the two-index anti-symmetric and fundamental representations. The "baryon" type denotes schematically where the singlet is to be found (including also the possibility of using the charge conjugates). Note that, because of $\epsilon^{abcde}$, the last model admits baryons of both types.

| Name | Gauge group | $\psi$ | $\chi$ | Baryon type |
|------|-------------|--------|--------|-------------|
| M1 | $SO(7)$ | $5 \times \mathbf{F}$ | $6 \times \mathbf{Spin}$ | $\psi\chi\chi$ |
| M2 | $SO(9)$ | $5 \times \mathbf{F}$ | $6 \times \mathbf{Spin}$ | $\psi\chi\chi$ |
| M3 | $SO(7)$ | $5 \times \mathbf{Spin}$ | $6 \times \mathbf{F}$ | $\psi\psi\chi$ |
| M4 | $SO(9)$ | $5 \times \mathbf{Spin}$ | $6 \times \mathbf{F}$ | $\psi\psi\chi$ |
| M5 | $Sp(4)$ | $5 \times \mathbf{A}_2$ | $6 \times \mathbf{F}$ | $\psi\chi\chi$ |
| M6 | $SU(4)$ | $5 \times \mathbf{A}_2$ | $3 \times (\mathbf{F}, \overline{\mathbf{F}})$ | $\psi\chi\chi$ |
| M7 | $SO(10)$ | $5 \times \mathbf{F}$ | $3 \times (\mathbf{Spin}, \overline{\mathbf{Spin}})$ | $\psi\chi\chi$ |
| M8 | $Sp(4)$ | $4 \times \mathbf{F}$ | $6 \times \mathbf{A}_2$ | $\psi\psi\chi$ |
| M9 | $SO(11)$ | $4 \times \mathbf{Spin}$ | $6 \times \mathbf{F}$ | $\psi\psi\chi$ |
| M10 | $SO(10)$ | $4 \times (\mathbf{Spin}, \overline{\mathbf{Spin}})$ | $6 \times \mathbf{F}$ | $\psi\psi\chi$ |
| M11 | $SU(4)$ | $4 \times (\mathbf{F}, \overline{\mathbf{F}})$ | $6 \times \mathbf{A}_2$ | $\psi\psi\chi$ |
| M12 | $SU(5)$ | $4 \times (\mathbf{F}, \overline{\mathbf{F}})$ | $3 \times (\mathbf{A}_2, \overline{\mathbf{A}_2})$ | $\psi\psi\chi, \psi\chi\chi$ |

Note that these models are not expected to be in the conformal window, but the logic is that they could be brought into it e.g. by the addition of extra matter that decouples at the $\Lambda_{\mathrm{HC}}$ scale, thus fulfilling the expectations discussed in the introduction. However, the one-loop $\gamma$-function does not depend on the number of fermions in a given irrep so the $\gamma$-function we compute will be the same as that of the corresponding conformal theory. It is on these conformal models that we need to focus first, searching for those giving rise to the most negative anomalous dimensions. As a second step, one should check that it is possible to reach a confining phase, by giving mass to some of the fermions, while still maintaining enough light fermions for a phenomenologically acceptable pattern of symmetry breaking.

There is a potential confusion in the number of entries of Table 2, e.g. (10) gives only one result for the $(1/2, 1)$ operator while Table 2 has two values. This is so because there are two inequivalent ways of assigning $X, Y, Z$ to the actual fermionic content of the theory. Denoting by $\psi$ and $\chi$ the fermions of a specific model, $\psi_\alpha \psi^\dagger_{(\dot\alpha} \chi^\dagger_{\dot\beta)}$ and $\chi_\alpha \psi^\dagger_{(\dot\alpha} \psi^\dagger_{\dot\beta)}$ renormalize differently.

Moreover, not all $(1/2, 0)$ represent potential top-partners. Depending on the assignment of SM charges to the hyper-quarks, some of them may give rise to the wrong irrep for the bound state, e.g. a **6** of color $SU(3)$. We do not repeat the details of the assignment of SM charges to the hyper-quarks for these models, that can be found in [7]. An example of a fully worked out list of bound states and their quantum numbers can be found in [4] for M8 and [6] for M6. Similar considerations for each model lead to Table 2.

The next step is to estimate the position of the fixed point for theories neighboring M1...M12 and to evaluate the $\gamma$-function at the critical value of the coupling to obtain the anomalous dimensions. As stressed in the Introduction, this is impossible to do rigorously within perturbation theory. To begin with, the $\beta$-function beyond two-loop is scheme dependent and so is the value of the coupling at the fixed-point. Since both the $\beta$ and $\gamma$-functions are computed in the $\overline{\mathrm{MS}}$ scheme, we obviously restrict ourselves to that. The *existence* of the fixed-point is

Table 2: Coefficient $A$ of the $\gamma$-function according to eq. (11)

|  | potential top-partners $(1/2,0)$ | other $(1/2,0)$ | $(1/2,1)$ | $(3/2,0)$ | $\overset{(\sim)}{\psi}\psi$ | $\overset{(\sim)}{\chi}\chi$ |
|---|---|---|---|---|---|---|
| M1 | -27/8,  - 9/2 | -39/8 | 9/8,  3/2 | 33/8 | -9 | -63/8 |
| M2 | -11/2,  -6 | -15/2 | 5/2,  2 | 13/2 | -12 | -27/2 |
| M3 | -39/8,  -9/2,  -27/8 |  | 9/8,  3/2 | 33/8 | -63/8 | -9 |
| M4 | -11/2,  -6,  -15/2 |  | 5/2,  2 | 13/2 | -27/2 | -12 |
| M5 | -3/2,  -6 | -15/2 | 1/2,  2 | 9/2 | -12 | -15/2 |
| M6 | -15/4,   -15/2 | -35/4 | 5/4,  5/2 | 25/4 | -15 | -45/4 |
| M7 | -45/8,  -27/4 | -81/8 | 27/8,  9/4 | 63/8 | -27/2 | -135/8 |
| M8 | -15/2,  -6,  -3/2 |  | 1/2,  2 | 9/2 | -15/2 | -12 |
| M9 | -45/8,  -15/2 | -105/8 | 35/8,  5/2 | 75/8 | -165/8 | -15 |
| M10 | -45/8,  -27/4,   -81/8 |  | 27/8,  9/4 | 63/8 | -135/8 | -27/2 |
| M11 | -35/4,  -15/2,  -15/4 |  | 5/4,  5/2 | 25/4 | -45/4 | -15 |
| M12 | -66/5,  -54/5,   -18/5 |  | 6/5,  18/5 | 42/5 | -72/5 | -108/5 |
|  | -24/5,  -36/5 | -72/5 | 24/5  12/5 | 48/5 | -108/5 | -72/5 |

however a universal property, albeit not accessible from perturbation theory unless one goes to the case of parametrically small coupling as in [33].

One could try to look at QCD, (defined here as a $SU(3)$ gauge theory with $N_f$ massless Dirac fermions[6]) for guidance, but even in this much studied case the situation is still unclear. Hoping not to misrepresent or neglect too many of the lattice results, reviewed in [34, 35], it seems that the conformal window should start from $N_f$ somewhere in the range 8-12 with $N_f = 8$ likely to be outside [36, 37] (thus chirally broken and confining). While [38–43] find $N_f = 12$ conformal, [44, 45] find results compatible with the chirally broken phase. The intermediate situation $N_f = 10$ (lattice computations are more easily performed with even numbers of flavors) is even more unclear [46–49].

Of course, science should not be done by consensus but by actual computations and experiments, so hopefully these disagreements will be resolved by the lattice community. However, given the limitedness of the scope of this discussion and the impossibility for us to make an educated judgment on the controversial lattice results, let us consider the majority opinion on these matters and assume that $N_f = 8$ is confining and $N_f = 12$ conformal. One can then ask the naive question of what are the perturbative predictions at various loop orders. Amusingly, it is the two-loop $\beta$-function whose predictions agree best with the above assumption as can be seen in Table 3. The three and four-loop results seem to overestimate the size of the conformal window, finding zeros for $N_f = 7$ and 8 respectively, while adding the five-loop result changes the picture completely putting $N_f = 12$ outside the conformal window.

We see that for high values of $N_f$ (near the perturbative edge of the conformal window at $N_f = 16.5$) the solution is small and stable, as expected. For smaller values of $N_f$ however, it is not clear at what loop order the improvement stops and the very different behavior of the five-loop solution weakens the results of [31], (obtained before the five-loop result [50] was published), where the good agreement between the three and four-loop result was used to argue about the validity of the perturbation theory even for $N_f \approx 8$.

Given how uncertain the situation is in the QCD case, we have little hope to make more quantitatively sound statements in our case. We will assume the following heuristic criteria

---

[6]When discussing QCD it is customary to count the number of Dirac fermions $N_f$ and we abide by this convention. In the rest of the paper however, we always count Weyl spinors, so for instance in Figs. 3 and 4 $N_F$ denotes the number of Weyl spinors in the fundamental irrep. Thus, if comparing, keep in mind that $N_F = 2N_f$.

Table 3: Values of the critical coupling $\alpha^*$ obeying $\beta(\alpha^*) = 0$ in the scheme of [50] for different loop order $L$ and number of Dirac flavors $N_f$ in the $SU(3)$ hyper-color theory. The / denotes the absence of a solution.

| | $N_f$ | | | | | | | | | |
| | 7 | 8 | 9 | 10 | 11 | 12 | 13 | 14 | 15 | 16 |
|---|---|---|---|---|---|---|---|---|---|---|
| $L = 2$ | / | / | 5.24 | 2.21 | 1.23 | 0.754 | 0.468 | 0.278 | 0.143 | 0.0416 |
| $L = 3$ | 2.46 | 1.46 | 1.03 | 0.764 | 0.579 | 0.435 | 0.317 | 0.215 | 0.123 | 0.0397 |
| $L = 4$ | / | 1.55 | 1.07 | 0.815 | 0.626 | 0.470 | 0.337 | 0.224 | 0.126 | 0.0398 |
| $L = 5$ | / | / | / | / | / | / | 0.406 | 0.233 | 0.127 | 0.0398 |

for the existence of a fixed point in our models, namely i) that the fixed point exists at all loop expansions available and ii) that the value of the anomalous dimension does not exceed the unitarity bound $\gamma^* \geq -3$ for these operators. ($9/2 - 3 = 3/2$, $9/2$ being the classical dimension of the operators and $3/2$ the unitarity bound.)

For these models we observe a similar trend as for QCD, namely that the three and four-loop $\beta$-functions give rise to a larger conformal region, thus the above conditions are dominated by the two-loop results. Clearly, inserting the higher-loop values for $g^*$ into the one-loop expression (11) is not a consistent approximation, however we prefer to present the results this way other than just giving the critical value of $g^*$ since the anomalous dimension has a more physical interpretation and is less scheme dependent.

In Figures 3,4 we show the models of Table 2 and the neighboring models obtained by increasing the amount of matter. Each model is represented by a circle. The models with matter content as in Table 1 are always located at the lowest left corner and the numbers on the axis denote the number of Weyl spinors. If there is no solution for the conditions i) or ii) above, the model is regarded to be confining and is represented by a yellowish circle. If both conditions are obeyed, the theory is considered to be conformal and we present the largest and lowest value for $\gamma^*$ obtained replacing the solution to $\beta(g^*) = 0$ at 2,3,4 loop into (11) where $A$ is chosen from Table 2 to be the largest one in absolute value among those of potential top-partners.

The red dashed curve indicates the "conformal house" [21] prescription $11 l_2(\mathbf{Ad}) - 4(N_\psi l_2(\psi) + N_\chi l_2(\chi)) < 0$.

One can then ask the question of how the anomalous dimensions of the QCD-like model behave under similar assumptions. Here we have the luxury of having the expression of $\gamma$ up to three loops and thus we can perform a more refined analysis by inserting the zero of the $L + 1$ loop $\beta$-function into the $L$ loop $\gamma$. Two operators, related to the proton, are considered in the literature, with $\gamma$-functions denoted by $\gamma_+$ and $\gamma_-$. Their values coincide at one-loop. We find the values for $L = 1, 2, 3$ displayed in Table 4 and 5 . The values of the last line of these tables agrees with [31] after multiplying by the factor of 2 dicussed in footnote 4.

The largest (negative) values for the anomalous dimensions are always arising by using the zeros of the two-loop $\beta$-function, but we argued that this may not be a drawback near the non-perturbative edge of the conformal window.

## 4 Conclusions

In this work we discussed various issues of relevance to gauge theories of Partial Composite-ness. First, we revisited and extended the computation of [18] of the anomalous dimension

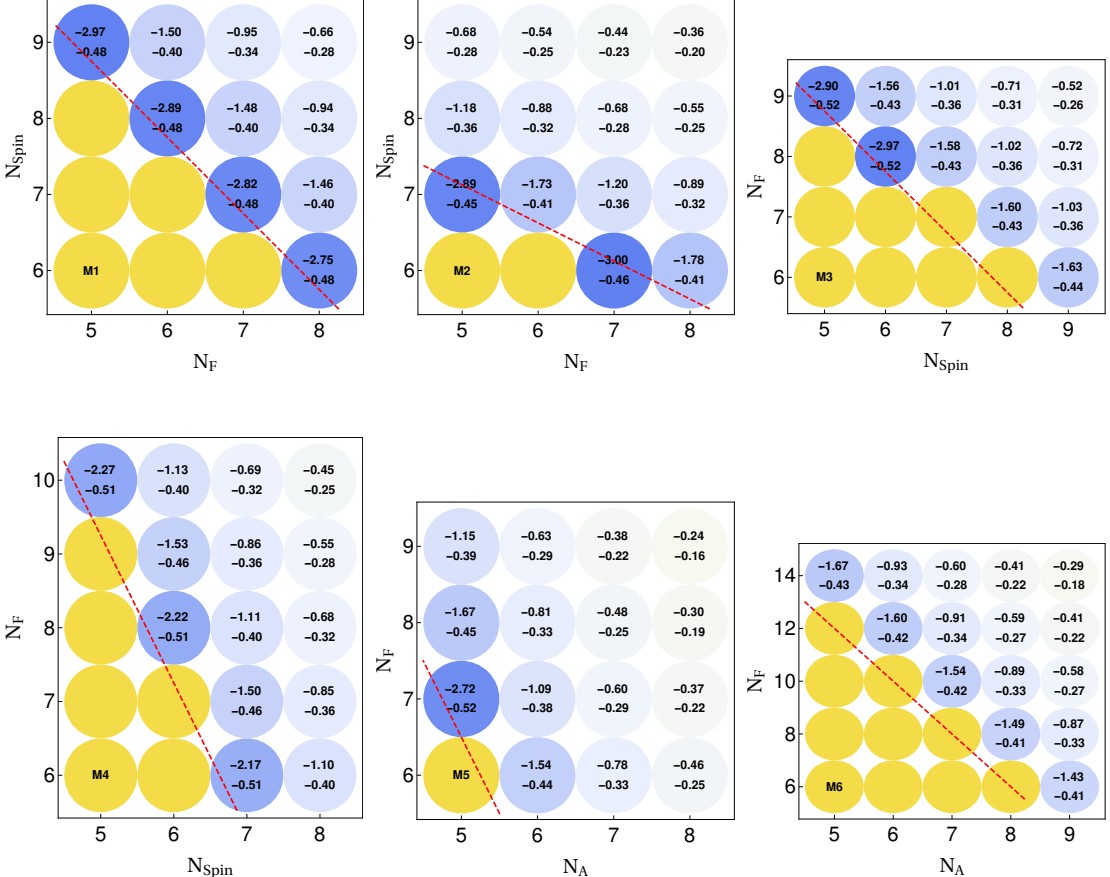

Figure 3: Models M1 to M6 and their neighbours with $N_X$ representing the number of Weyl fermions in the $X$ representation. Yellow circles represent potentially *confining* models whereas blue circles represent models likely to be in the *conformal* window, with the estimated maximal and minimal value of $\gamma^*$ displayed. Our heuristic arguments for this classification are described in the text. The red dashed curve indicates the "conformal house" [21] prescription.

of generic fermionic trilinears. We showed that all operators of higher spin acquire a positive anomalous dimension ($A > 0$ in Table 2) and thus decouple from the theory even more than they would already do classically. On the other hand, the potential partners all have a negative anomalous dimension and from the very rough estimate of the location of the fixed point it does not seem unlikely that there be cases where $\gamma^* \approx -2$.

The location of the most promising theories can be read off from Fig. 3 and 4 as the locations of the "darkest" points, where the range of possible $\gamma^*$ values, with our heuristics, stretches past $-2$. As long as the confining theory is above or to the right of one of the models of Table 1 it is possible to leave the conformal region by giving mass to some fermions but retaining enough light ones to yield an acceptable phenomenology.

The following observations however, mitigate the above results. First of all, in full generality, one of the two fermionic bilinears always have a one-loop anomalous dimension which is larger (in absolute value) than that of all the fermionic trilinears. (This was observed in the QCD context in [31].) This is a potential problem for these models *unless* the expressions for the higher-loop $\gamma$-functions cross at some point (as they actually do perturbatively in QCD). The reason is that we need $\gamma^* \approx -2$ (even assuming a better overlap coefficient than that of

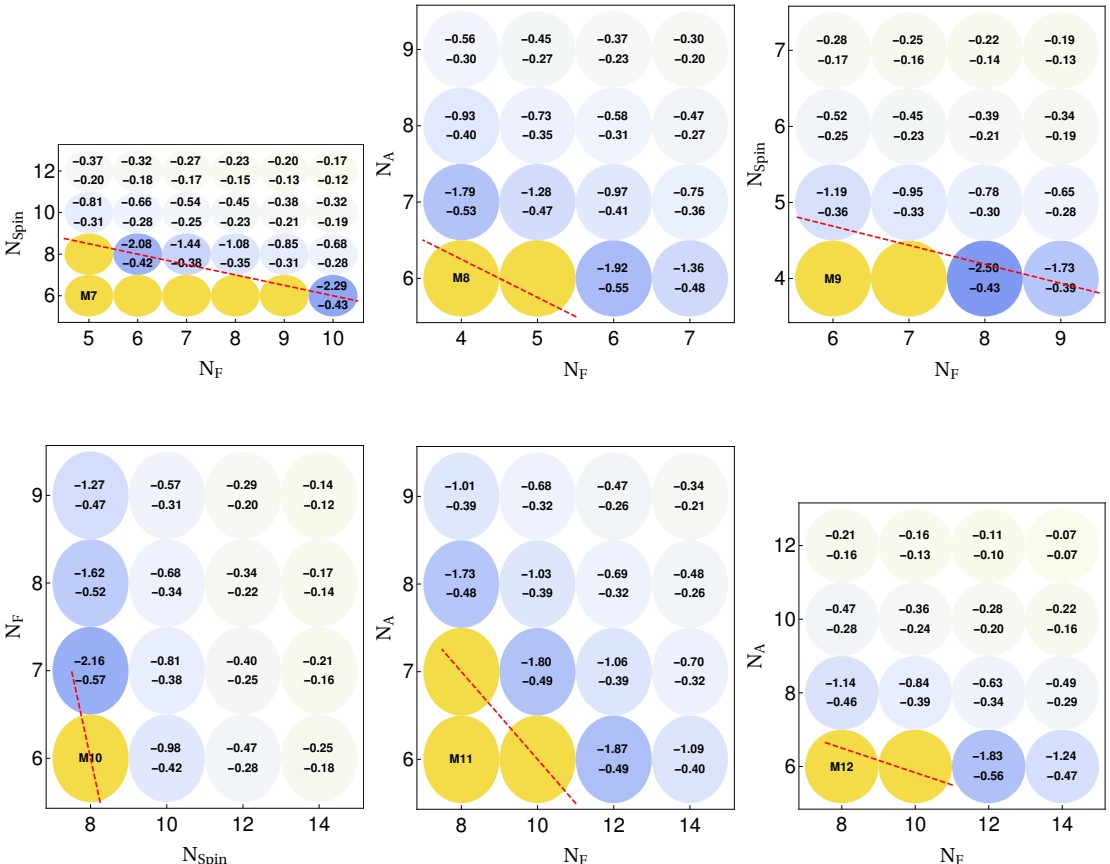

Figure 4: Models M7 to M12 and their neighbours with $N_X$ representing the number of Weyl fermions in the $X$ representation. Yellow circles represent potentially *confining* models whereas blue circles represent models likely to be in the *conformal* window, with the estimated maximal and minimal value of $\gamma^*$ displayed. Our heuristic arguments for this classification are described in the text. The red dashed curve indicates the "conformal house" [21] prescription.

M6 [12]) in order for the corresponding fermionic operator $\mathcal{O}$ to become a viable top-partner[7] but if the bilinears acquire a similar anomalous dimension they would approach the free field limit (3-2=1) where the bootstrap argument of [51] shows that fine-tuning is reintroduced.

Ironically, the most studied models M6 and M8 and the QCD-like one are among those for which we do not find $\gamma^* \lesssim -2$ solutions.

A second curious fact is that, among the spin 1/2 composite operators, it is often those that do *not* qualify as top-partners (typically QCD sextets) that acquire the largest anomalous dimension (cfr $A$ in Table 2 in the first two columns). There is nothing directly wrong with this fact, but it shows that in some models the top-partners do not stand out as those with the leading anomalous dimensions among all the spin 1/2 operators.

A two-loop computation of the anomalous dimensions for these objects would be interesting, if only to see if the above trends continue. It is reasonable to expect, comparing with the QCD results [28] [32], that the two-loop $\gamma$-function for the top-partners has the same sign as the one-loop one, helping making the partners anomalous dimensions more negative for the same value of the critical coupling.

As hopefully we made clear in the main text, while the computation of the $\gamma$-function

---

[7]We need $9/2 - 2 = 5/2$, so that the linear coupling $\mathcal{O}t$ in the Lagrangian becomes marginal, i.e. of dimension 4.

Table 4: Anomalous dimension $\gamma_+^*$ for the QCD-like model at $L$ loops obtained inserting the $L + 1$ loop critical coupling $\alpha^*$.

| | $N_f$ | | | | | | | |
| --- | --- | --- | --- | --- | --- | --- | --- | --- |
| | 9 | 10 | 11 | 12 | 13 | 14 | 15 | 16 |
| $L = 1$ | -1.67 | -0.703 | -0.393 | -0.240 | -0.149 | -0.0885 | 0.0455 | -0.0132 |
| $L = 2$ | -0.385 | -0.277 | -0.204 | -0.150 | -0.107 | -0.0715 | -0.0404 | -0.0128 |
| $L = 3$ | -0.0150 | -0.108 | -0.128 | -0.119 | -0.0969 | -0.0688 | -0.0400 | -0.0128 |

Table 5: Anomalous dimension $\gamma_-^*$ for the QCD-like model at $L$ loops obtained inserting the $L + 1$ loop critical coupling $\alpha^*$.

| | $N_f$ | | | | | | | |
| --- | --- | --- | --- | --- | --- | --- | --- | --- |
| | 9 | 10 | 11 | 12 | 13 | 14 | 15 | 16 |
| $L = 1$ | -1.67 | -0.703 | -0.393 | -0.240 | -0.149 | -0.0885 | 0.0455 | -0.0132 |
| $L = 2$ | -0.474 | -0.326 | -0.233 | -0.166 | -0.116 | -0.0753 | -0.0416 | -0.0129 |
| $L = 3$ | -0.110 | -0.163 | -0.160 | -0.138 | -0.106 | -0.0730 | -0.0413 | -0.0129 |

stands on firm footing, the estimate of the anomalous dimension $\gamma^*$ involves a fair amount of assumptions and speculations. We see no harm in doing this as long as we only use them as a guidance. However, by themselves, these perturbative computations cannot be taken as a proof (or a disproof) of any statement about the validity of these models.

A last subject discussed in this paper, confined to the Appendix but of broader interest than just to Partial Compositeness, is the computation of the group theory factors that enter in the expression of the four-loop $\beta$-function in multi-fermions theories [20]. Here we present practical formulas and numerical results, a few of them new to our knowledge, to facilitate working with fourth-order Casimir operators, their corresponding invariant tensors and the products of such tensors between different irreps.

## Acknowledgments

The authors are supported by the Knut and Alice Wallenberg foundation under the grant KAW 2017.0100 (SHIFT project). We would like to thank T. DeGrand and Y. Shamir for comments of the manuscript and J.A. Gracey, C. Pica and F. Sannino for email exchanges.

## A  Useful tables of fourth-order invariants

In this appendix we collect a few results on fourth order indices for simple Lie algebras that are useful for higher loop computations, independently on the applications to partial compositeness.

For any simple Lie algebra [8] $\mathcal{L}$ it is always possible, convenient and sufficiently general to chose the generators $T^a$ in an arbitrary irrep $R$ to be orthogonal and uniformly normalized, that is: $\mathrm{tr}(T^a T^b) = l_2(R)\delta^{ab}$. $l_2(R)$ is known as the quadratic index of the irrep $R$. Choosing the

---

[8]We use the "physicist" convention and denote $\mathcal{L}$ by the corresponding group $G = SU(n), SO(n)\dots$.

normalization of one (typically the fundamental **F**) irrep fixes all the normalizations. Physicists usually assume $l_2(\mathbf{F}) = 1/2$, while mathematicians prefer $l_2(\mathbf{F}) = 1$. In the appendix we choose $l_2(\mathbf{F}) = 1$ commenting, where necessary, on how to revert to $l_2(\mathbf{F}) = 1/2$ to comply with the QFT literature. Having chosen the invariant tensor $\delta^{ab}$ allows us not to distinguish between raised and lowered adjoint indices.

We can also define the quadratic Casimir operator as $C_2(R)\mathbf{1} = \delta^{ab}T^a T^b$, which is proportional to the identity for any irrep. Taking the trace implies the consistency condition $C_2(R)\dim(R) = l_2(R)\dim(G)$. Since the condition is valid for $R = \mathbf{F}$ as well, $l_2(R) = C_2(R)\dim(R)/(C_2(\mathbf{F})\dim(\mathbf{F}))$. The quadratic index and Casimir are thus simply related to each other.

One can define higher invariants in a similar way. The cubic index (known in physics as the anomaly coefficient) is defined by $\frac{1}{2}\mathrm{tr}(T^a T^b T^c + T^a T^c T^b) = l_3(R)\delta^{abc}$, with $\delta^{abc}$ a manifestly fully symmetric and traceless tensor that is non-zero only for $SU(n \geq 3)$. Now one usually sets $l_3(\mathbf{F}) = 1$ to define the overall normalization of $\delta^{abc}$ and uses it to define the cubic Casimir $C_3(R)\mathbf{1} = \delta^{abc}T^a T^b T^c$ for any irrep. Once again the consistency condition implies, with our normalization, $l_3(R) = C_3(R)\dim(R)/(C_3(\mathbf{F})\dim(\mathbf{F}))$ when these quantities are non-zero. Note that, even after choosing $l_3(\mathbf{F}) = 1$, the tensor $\delta^{abc}$ is still implicitly dependent on how we normalized the generators by choosing $l_2(\mathbf{F})$, and a similar argument applies to higher tensors.

The values of the quadratic and cubic indices or Casimirs are well known in the literature, e.g. [52–54] and will not be reviewed here.

To construct the quartic (and higher) invariant we need to take care of an additional subtlety, since the fully symmetric tensor

$$d^{abcd}(R) = \frac{1}{6}\mathrm{tr}(T^a T^b T^c T^d + T^a T^d T^b T^c + T^a T^c T^d T^b + T^a T^c T^b T^d + T^a T^d T^c T^b + T^a T^b T^d T^c) \tag{12}$$

is not irreducible anymore and so it is not the same, up to a proportionality constant, for every irrep. This can be easily fixed, for all algebras other than $SO(8)$, by constructing the traceless component

$$l_4(R)\delta^{abcd} = d^{abcd}(R) - \kappa(R)\left(\delta^{ab}\delta^{cd} + \delta^{ac}\delta^{bd} + \delta^{ad}\delta^{bc}\right), \tag{13}$$

where

$$\kappa(R) = \frac{l_2(R)^2 \dim(G)/\dim(R) - l_2(R)l_2(\mathbf{Ad})/6}{\dim(G) + 2} \tag{14}$$

and $\delta^{abcd}$ is defined up to a proportionality constant that can be fixed by taking $l_4(\mathbf{F}) = 1$ just as in the previous case. Moreover, setting $C_4(R)\mathbf{1} = \delta^{abcd}T^a T^b T^c T^d$ yields $l_4(R) = C_4(R)\dim(R)/(C_4(\mathbf{F})\dim(\mathbf{F}))$.

The case of $SO(8)$ is special because there exist another quartic invariant symmetric traceless tensor $e^{abcd}$ constructed using the anti-symmetric $\epsilon^{\mu_1 \cdots \mu_8}$ tensor, treating $a, b \cdots = 1, 2 \ldots 28$ as multi-indices $[\mu, \nu] = [1,2], [1,3] \ldots [7,8]$, e.g. $e^{1,14,23,28} = \epsilon^{1,2,3,4,5,6,7,8} = 1$. This component does not affect the tensor irreps but for the spinor we have, for $SO(8)$ only

$$-\frac{1}{2}\delta^{abcd} = d^{abcd}(\mathbf{Spin}) - \frac{1}{12}\left(\delta^{ab}\delta^{cd} + \delta^{ac}\delta^{bd} + \delta^{ad}\delta^{bc}\right) + \frac{1}{8}e^{abcd}, \tag{15}$$

in other words, $l_4(\mathbf{Spin}) = -1/2$. Note that $\delta^{abcd}e^{abcd} = 0$.

The value of $l_4$ can be extracted from the work of [55–57] [9]. We present them in Table 6. Note that the quartic index is zero for $SU(2), SU(3)$ and all exceptional algebras.

---

[9]We warn the reader that the literature uses varying notations and conventions. In particular, the indices are not those originally defined in [58] but are instead related to the "modified" ones in [56]. (To be more precise, they are proportional to the object $D^{(4)}(R)$.) The Casimir in [56] is denoted by $J_4(R)$ and is related to $C_4(R)$ by an overall $R$ independent proportionality constant.

Table 6: Quartic indices of the commonest irreps. The symbol $\lfloor x \rfloor$ denotes the floor of $x$.

|  | F | $\mathbf{A_2}$ | $\mathbf{S_2}$ | Ad | Spin |
|---|---|---|---|---|---|
| $SU(n \geq 4)$ | 1 | $n-8$ | $n+8$ | $2n$ | $\times$ |
| $Sp(2n \geq 4)$ | 1 | $2n-8$ | $\times$ | $2n+8$ | $\times$ |
| $SO(n \geq 7)$ | 1 | $\times$ | $n+8$ | $n-8$ | $-2^{\lfloor (n-9)/2 \rfloor}$ |

Table 7: The numerical values of $d^{abcd}(R_1)d^{abcd}(R_2)$ for $SU(2,3,4,5,6)$ respectively, with the choice $l_2(\mathbf{F})=1$. For the "physicist" normalization $l_2(\mathbf{F})=1/2$ multiply each value by $1/16$.

|  | F | $\mathbf{A_2}$ | $\mathbf{S_2}$ | Ad |
|---|---|---|---|---|
| **F** | 5/4, 20/3, 445/16, 1972/25, 2135/12 | 0, 20/3, -5/2, -276/25, 140/3 | 20, 340/3, 885/2, 31276/25, 8680/3 | 20, 120, 440, 1240, 2940 |
| $\mathbf{A_2}$ |  | 0, 20/3, 340, 30708/25, 8960/3 | 0, 340/3, 300, 26292/25, 11200/3 | 0, 120, 640, 2280, 6720 |
| $\mathbf{S_2}$ |  |  | 320, 5780/3, 7380, 526708/25, 150080/3 | 320, 2040, 7680, 22120, 53760 |
| **Ad** |  |  |  | 320, 2160, 8320, 24400, 60480 |

Using the above formulas it is straightforward to evaluate the products $d^{abcd}(R_1)d^{abcd}(R_2)$ arising in the four-loop $\beta$-function [20]. The general expression for $\delta^{abcd}\delta^{abcd}$ can be obtained by brute force or by doing some "reverse engineering" on the formulas in [59] for $d^{abcd}(\mathbf{F})d^{abcd}(\mathbf{F})$ and are given by (recall that we normalize to $l_2(\mathbf{F})=1$ here)

$$SU(n): \quad \delta^{abcd}\delta^{abcd} = \frac{(n^2-1)(n^2-4)(n^2-9)}{6(n^2+1)}, \tag{16}$$

$$SO(n): \quad \delta^{abcd}\delta^{abcd} = \frac{n(n-3)(n^2-1)(n^2-4)}{48(n^2-n+4)}, \tag{17}$$

$$Sp(2n): \quad \delta^{abcd}\delta^{abcd} = \frac{n(2n+3)(n^2-1)(4n^2-1)}{12(2n^2+n+2)}. \tag{18}$$

From these expressions $d^{abcd}(R_1)d^{abcd}(R_2)$ can be derived as

$$d^{abcd}(R_1)d^{abcd}(R_2) = l_4(R_1)l_4(R_2)\delta^{abcd}\delta^{abcd} + \kappa(R_1)\kappa(R_2)3\dim(G)(\dim(G)+2) \tag{19}$$

adding a factor $\frac{1}{64}e^{abcd}e^{abcd} = 315/8$ in the $SO(8)$ case $R_1 = R_2 = \mathbf{Spin}$.

We present some numerical results explicitly in the following "multiplication tables" Table 7, Table 8 and Table 9 for the groups $SU$, $Sp$ and $SO$ respectively. Note that there are non-zero entries for $SU(2)$ and $SU(3)$ as well, since we are dealing with the reducible tensor. In some cases there is some redundancy, since e.g. for $SU(3)$ $\mathbf{A_2} = \overline{\mathbf{F}}$.

Table 8: The numerical values of $d^{abcd}(R_1)d^{abcd}(R_2)$ for $Sp(4,6,8)$ respectively, with the choice $l_2(\mathbf{F}) = 1$. For the "physicist" normalization $l_2(\mathbf{F}) = 1/2$ multiply each value by 1/16.

|  | **F** | **A$_2$** | **Ad** |
|---|---|---|---|
| **F** | 10, 161/4, 114 | 5, 56, 306 | 165, 700, 2130 |
| **A$_2$** |  | 160, 1064, 4104 | 240, 1960, 9000 |
| **Ad** |  |  | 2880, 13160, 43080 |

Table 9: The numerical values of $d^{abcd}(R_1)d^{abcd}(R_2)$ for $SO(7,8,9,10,11)$ respectively, with the choice $l_2(\mathbf{F}) = 1$. For the "physicist" normalization $l_2(\mathbf{F}) = 1/2$ multiply each value by 1/16.

|  | **F** | **S$_2$** | **Ad** | **Spin** |
|---|---|---|---|---|
| **F** | 161/4, 70, 114, 705/4, 1045/4 | 2961/4, 1330, 2244, 3600, 22165/4 | 385/4, 210, 420, 780, 5445/4 | -49/16, -35/4, -75/2, -555/8, -935/4 |
| **S$_2$** |  | 58401/4, 27160, 47454, 78840, 502645/4 | 11025/4, 5880, 11550, 21240, 148005/4 | 1071/16, 70, 165/2, -90, -3575/4 |
| **Ad** |  |  | 4865/4, 2520, 4830, 8760, 60885/4 | 1855/16, 210, 1365/2, 1020, 11385/4 |
| **Spin** |  |  |  | 1001/64, 70, 435/2, 5745/16, 8965/4 |

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
