# Peer review of "Anomalous dimensions of potential top-partners"

_SciPost Physics, doi:SciPost Phys. 7, 027 (2019)_

## Round 1 · Referee Report · Anonymous (Referee 1) · 2019-5-30

Strengths

1- The paper provides comprehensive tabulation of analytic results for each candidate BSM model, which in particular can help guide future lattice work.

2- The appendix collects useful details on fourth-order group theoretic invariants which are of general interest.

3- Generally speaking the paper is clearly written and methodological.

Weaknesses

1- The 2,3, and 4 loop estimates for the value $g^*$ of the infrared fixed point are typically widely different, leading to corresponding large uncertainties in the estimated value of $\gamma^*$.

Report

This paper deals with a class of strongly coupled asymptotically free theories, which are candidates for beyond the Standard Model (BSM) physics, containing a composite Higgs as well as a partially composite top quark. Specifically, the paper attempts to provide analytic estimates for the anomalous dimension of top partner baryonic states of the new strong interaction.

This is a timely paper, which can among other things
help guide further lattice studies of similar models.
The main limitation is that the the different perturbative based estimates for the anomalous dimensions often vary significantly. But, with the authors, I agree that given the amount of work involved in a lattice study, any hints coming from analytic calculations are helpful.

I recommend the paper for publication once the authors have considered the optional changes listed below.

Requested changes

1- The logic of the calculation could be explained a bit better, preferably already in the introduction. Specifically, the following sentences may confuse the reader. In the conclusion:

'The location of the most promising theories can be read off from Fig. 3 and 4 as the locations of the "darkest" points, where the range of possible $\gamma^*$ values, with out heuristics, stretches past $-2$.'

Of course, all candidate BSM theories must ultimately be confining, and not conformal. Thus, what the authors really mean here is to pick a yellow-circle model that is closest to one of those "darkest" circles.

("with out heuristics" appears to have some typo, I didn't understand what the authors meant.)

Similarly, on page 8, they say:

'Note that these models are not expected to be in the conformal window, but the logic is that they could be brought into it e.g. by the addition of extra matter that decouples at the HC scale.'

Once again, what is missing is to spell out, explicitly, that ultimately the actual BSM models must remain confining, that is, outside of the conformal window. Thus, the most promising models (from this perspective) could be those that lie just outside the conformal window.

2- In the discussion of QCD (page 9) the authors use the familiar notation $N_f$ for the number of Dirac fermions. By contrast, in Figures 3,4 they use $N_F$ which counts the number of Weyl fermions in representations $F$ and $\bar{F}$ together. Hence, in fact, $N_F = 2 N_f$. Clarifying this may help avoid confusion for the reader!

3- Questions related to Figs 3,4: a) Is $g^*$(2 loops) always/typically larger than $g^*$(3,4 loops)? b) Related, does the larger $|\gamma^*|$ always come from 2 loops?

4- Page 3, bottom line, the authors discuss the 4-index objects $Df$ and $D\bar{f}$. But $f$ and $\bar{f}$ each carry only two indices. The other two indices (one dotted and one undotted) come from $D$. The authors can clarify this.

5- In the conclusion, the authors have the parenthetical equation $9/2-2=5/2$. It might help the reader to remind why 5/2 is the "best" value of the (quantum) dimension of the top-partner operator.

  • validity: high
  • significance: good
  • originality: good
  • clarity: high
  • formatting: -
  • grammar: -

Author:  Gabriele Ferretti  on 2019-06-27  [id 550]

(in reply to Report 1 on 2019-05-30)

Thank you for your comments and your suggestions!

1) We tried to expand and clarify the argument, starting with the third paragraph in the introduction and adding further comments in the text where you pointed to.

As for the comment about confining vs. conformal, this is also related to a comment of the other Referee, who, "seems" to be saying the opposite of what you are saying. We believe however that you are both thinking the same thing, namely: the IR theory must ultimately be confining (obviously) but, for the purpose of finding a large anomalous dimensions for the top-partners, it could very well start inside the conformal window and be taken away the fixed point by a relevant deformation, such as a fermion mass.

What is important is that, after this happens, there are enough light fermions left to guarantee a sensible phenomenology. The interest is thus to look at confining models adjacent to conformal models with large anomalous dimensions. The actual number of dynamical fermions, and thus whether the model is in the conformal window or not, depends on the relation between the fermion masses and the energy scale.

Sorry about the typo: "out heuristics" should be "our heuristics".

2) Done. (In a footnote).

3) The answer to both questions is yes. We added a comment about it.

4) Done. (See also reply to point 2 of the other Referee)

5) Done. (In a footnote).

Author:  Gabriele Ferretti  on 2019-07-13  [id 558]

(in reply to Gabriele Ferretti on 2019-06-27 [id 550])
Category:
remark

It was pointed out to me by the other Referee that the revised version was not visible in the reply.
Sorry about this. It can be downloaded directly from the arXiv as the latest (V2) version.
https://arxiv.org/pdf/1905.08273.pdf
Best regards

---

## Round 1 · Referee Report · Luca Vecchi (Referee 2) · 2019-6-9

Report

Hi all,

The paper derives the 1-loop anomalous dimension of baryonic operators (of spin 1/2 and 3/2) in a class of 4D gauge theories previously proposed as possible UV realizations of composite Higgs models implementing partial compositeness. Using the 2,3,4 loop beta function of the associated non-abelian gauge theory, the authors then estimate the size of the anomalous dimension at a putative IR fixed point of the theory.

The paper is well-written, contains useful new results, and deserves to be published. Before I can recommend it for publication, however, the authors should address the following questions/comments:

0) Let me start with a comment, triggered by the very first equation in the paper (eq. (1)). I find the result of [11] quite surprising. I followed their approach, which seems to be correct, but the final numerical result is unexpected, to say the least. NDA tells us that the baryon wave-function (in units of $f$), and as defined in [11], is of order $g_*$, that is the typical hadron coupling. This is perfectly consistent with QCD. But the result of [11] turns out to be off by a factor of $O(20)$ smaller than the NDA expectation. Why is that? A "large $N=4$" factor would not explain it. Either there is a technical problem in their numerical analysis, or some unknown dynamical reason. Any idea which one is it?

Let me now talk about this paper…

1) At the beginning of Sec 2 the authors discuss which operators mix under RG (please specify right at the onset that you adopt dim-reg). The results are correct, but presented in a way they seem a bit obscure. It would be better to state clearly that often the mixing is forbidden by the (unbroken) flavor symmetry. For example this is why the $(1/2,0)$ in $XY^\dagger Z^\dagger$ does not mix with the ones in $XYZ$.

2) I am a bit confused by the discussion of $DFX$. I think it can in general mix… However, the point is that that operator can be removed via a field re-definition and is therefore redundant, not important to the analysis. For example, using eq(6.6) of 1008.4884 one sees that $JDFX$ (with $J$ some fermionic current) can always be written as a 4-fermion interaction $JXYZ$ (up to total derivatives). Therefore it seems to me that $DFX$ can mix with $XYZ$, but the effect is already taken into account by the matrix $a^{IJ}$.

3) In the second paragraph of page 5, please state explicitly that $a_X=-2C_X$ assumes $\xi=1$. In general $a_X=-2C_X\xi$.

4) I have the feeling eq (2) should be multiplied by an overall minus… Am I right? You say $X_B=\sqrt{Z_X}X_R$ and $[XYZ]_R=Z X_RY_RZ_R$, with $Z$ the vertex correction. The relation between renormalized and bare operators is thus $[XYZ]_R=Z(Z_XZ_YZ_Z)^{-1/2}[XYZ]_B$. From this follows that $\gamma=-\frac{d}{d\ln\mu}\ln[O]_R$ is the opposite of (2). But then I do not understand why you recover earlier results… Either there is a typo in (2) or I am missing something. Please clarify.

5) After (8), and before (9), it would very much help the presentation to show what is $a^{IJ}$ in terms of $C_X,C_Y,C_Z$. I understand that this is straightforwardly derived from (9) and (2), but the equation I suggest would put to use the notation introduced earlier, namely $a^{IJ}$.

6) A quantitative discussion of the QCD case must be added somewhere. It would help the reader gauge the results presented, and also make the paper more comprehensive. The authors may opt to add a row in the tables and a new figure. Or they could simply add a paragraph somewhere in Sec 3 to show what is the value of the QCD $\gamma$ using the fixed points found at 2,3,4 loops. The fact that the paper does not compare QCD-like theories to the other models forces the reader to jump from one ref to another in order to have a better assessment of the status.

(It is true that the estimate I am referring to may be found somewhere else, say [29], but the authors also know that the are factors of 2 wrong in that ref. This would not make a fair comparison.)

Also, I very much appreciate the unbiased approach of the authors. For this reason, I invite them to enlarge their discussion including a brief assessment of QCD-like theories. If not there, the reader might be induced to think that only the theories proposed in [5] are viable. This is especially important because the authors’ work is considered as an important reference by some members of the lattice community. Thanks!

And please also clarify to the referee (and the other readers) that being INSIDE the conformal window is perfectly fine: in these models some relevant deformation (typically a fermion mass) can push us away from the fixed point at parametrically low scales, and trigger confinement (see 0806.1235, 1506.00623).

  • validity: -
  • significance: -
  • originality: -
  • clarity: -
  • formatting: -
  • grammar: -

Author:  Gabriele Ferretti  on 2019-06-27  [id 549]

(in reply to Report 2 by Luca Vecchi on 2019-06-09)

Thank you for your comments and your suggestions!

0) It will be interesting to see if the results of [11] are generic or specific of this model. We prefer not to guess an answer and wait for more data from the lattice.

1) We added a sentence at the beginning stating that we use dim reg and work in the Feynman gauge. As far as the mixing is concerned, we added the clarification in the third paragraph of that section.

2) Yes, this is what we meant to say, in a slightly different way. When decomposing $DFX$ in the various Lorentz irreps, some pieces do indeed mix, but they are the ones that, as you say, can be rewritten by using the equations of motion. If you look at the paper hep-th/0412029, this point of view is discussed extensively at the beginning of Section 3. We slightly modified the text, but, since we don't think there is any disagreement between us on the actual results, we would not want to dwell too much on it.

3) Done. (In a footnote).

4) The sign that is confusing you comes from the fact that in dim. reg. the counterterms contain the factor $\Gamma(2-d/2)/(M^2)^{2-d/2} \approx 2/\epsilon - 2 \log M$. Thus, taking $-d/d\log M$ is the same as taking $+$ the residue in $\epsilon$. We defined our $a$'s as $+$ the residues of the poles and thus it probably has the opposite sign of what you are familiar with.

Eq. (2) is correct as it stands with our notation.

5) Ok...

6) We agree with your philosophy on this matter. The models in [5] and that in [8] are on the same footing from the "group theory" point of view and should be judged at the dynamical level by the lattice or (daring to dream) the experiment.

We agree that there is an overall factor of 2 missing in the published literature, we comment on this in a footnote and in the text.

To address the comparison with QCD-like theories, we added tables 3, 4 and 5 and some text, enjoy!

As you can see, the largest (negative) anomalous dimension is -1.67, arising from the 2 loop beta-function. Better than the higher loop ones, but still a bit smaller then -2. Given the qualitative nature of this estimate, we do not want to reach a definite conclusion on the viability of these models, or the other ones like it M6 or M8, but we simply state this fact in a neutral way in the conclusions.

As for the comment about confining vs. conformal, this is also related to a comment of the other Referee, who, "seems" to be saying the opposite of what you are saying. We believe however that you are both thinking the same thing, namely: the IR theory must ultimately be confining (obviously) but, for the purpose of finding a large anomalous dimensions for the top-partners, it could very well start inside the conformal window and be taken away the fixed point by a relevant deformation, such as a fermion mass.

What is important is that, after this happens, there are enough light fermions left to guarantee a sensible phenomenology. The interest is thus to look at confining models adjacent to conformal models with large anomalous dimensions. The actual number of dynamical fermions, and thus whether the model is in the conformal window or not, depends on the relation between the masses and the energy scale.

We added a clarification of this point in the introduction (third paragraph) and in the places that the other Referee found confusing.

---

## Round 2 · Referee Report · Anonymous · 2019-8-11

Report

The authors have addressed all my comments and I recommend the paper for publication in SciPost.

There appears to be a typo in footnote 6:
if $N_f$ is the number of Dirac fermions and $N_F$ of Weyl fermions then $N_f=N_F/2$ (and not the other way around,
as the footnote now has).

---

## Round 2 · Author Response

Sorry it took so long, I though I had already resubmitted, but something went wrong.

---

## Round 2 · List of Changes

All the changes are listed and discussed in the replies to the Referees.
Structural changes: Added Table 3,4,5, equation (9) and two references.

---

## Editorial Decision

published